# From moral distress to resilient ethical climate among general practitioners: Fostering awareness. A qualitative pilot study

Raf Coremans[1], Anton Saerens[1]*, Jan De Lepeleire[1‡], Yvonne Denier[2‡]

**1** Academic Centre for General Practice, KU Leuven, Leuven, Belgium, **2** Department of Public Health and Primary Care, Centre for Biomedical Ethics and Law, KU Leuven, Leuven, Belgium

☯ These authors contributed equally to this work.
‡ JDL and YD also contributed equally to this work.
* anton.saerens@icloud.com

## Abstract

### Background

Moral distress in and ethical climate of health care institutions are highly intertwined subjects and have been linked to various quality of care indicators as well as job turnover intentions among health care professionals. Predominantly, both phenomena have been studied in intensive care, palliative and in-hospital settings. We aimed to explore the experience of moral distress by general practitioners (GPs), the role of ethical climate in GP moral distress and how ethical climate and moral distress can result in moral resilience in general practice.

### Methods and findings

Between April and October 2021, we interviewed 13 doctors active in general practice in Flanders, Belgium, through semi-structured interviews. Data were processed and analysed using the Qualitative Analysis Guide of Leuven (QUAGOL). Most GPs had ample experience with morally distressing situations. Causes, determinants, and consequences do not differ significantly from other care settings. Moral distress can arise from conflicting views of good care, communication problems, and impending harm to third parties. We detected determinants of moral distress on micro-, meso- and macrolevels. GPs associate moral distress with job turnover and emotional, physical, existential, and quality of care effects. Several malleable factors can contribute to resilient ethical climates. This requires acquisition of vocabulary, skills, and knowledge.

### Conclusions

Moral distress and ethical climate are important emerging themes for GPs. This research identifies determinants and effects of moral distress and ethical climate in primary care and could help GPs leverage moral distress experiences into morally resilient primary care through multiple suggested strategies.

**Data Availability Statement:** For legal and ethical reasons, data cannot be shared in an unrestricted manner. Participants did not give consent in terms

of publication of their full transcript. Public availability would compromise patient confidentiality and participant privacy. EC/OBC Research is willing to act as institutional contact and to respond to external requests for data access via obc@kuleuven.be. This decision was made in accordance with EC/OBC.

**Funding:** The author(s) received no specific funding for this work.

**Competing interests:** The authors have declared that no competing interests exist.

# Introduction

## Moral resilience through thriving ethical climates

**Definitions of moral distress and ethical climate.** Moral distress (MD) and ethical climate (EC) are important emerging themes in medical ethics.

Multiple concepts of MD exist [1]. Moral distress occurs when a healthcare professional cannot or does not act on his or her moral judgment(s) (or what he or she believes to be right in a particular situation) because of contextual or internal constraints [2]. In literature, we can find a number of examples of MD among doctors: responding to the aggressive treatment wishes of family members at the end of a patient's life; scarcity of materials, money or time/space for helping patients, having to care for more patients than is actually feasible [3–5].

The "ethical climate" concept was first introduced by Victor and Cullen in 1988 [6]. Their publication was groundbreaking mainly through highlighting the extrapersonal circumstances that influence ethical decision-making and behaviour within an organization [6]. Several concepts of ethical climate in care organisations have been developed subsequently, each receiving critique for being incomplete, and only highlighting a part of the whole organisational context impacting ethical behaviour, such as democratic dialogue, circumstances that are conducive to ethical reflection etc. [7–9].

Based on these critiques, we will further use the following, more complete concept of EC: the EC of an organisation includes the whole organisational context that influences decisions and actions with ethical connotations by its employees [10].

EC then, is a very broad phenomenon with complex generative mechanisms and not merely the presence of 'people working on an ethical climate'. It exists in every organization, whether people are aware of it or not. It is an emergent phenomenon arising from and influenced by an organization's individuals and their interactions.

The organization's EC then, logically, is a major contextual or meso-level determinant in the development of a health professional's MD. The ethical climate, among other things, determines what expectations are set for ethical behaviour and thus influences ethical decision making. More concretely, if a health professional's conviction of doing the right thing conflicts with the expectations in the organization, this person will experience moral distress. An extreme example would be a 'progressive' physician working in a 'conservative' institution. Abortion or euthanasia are not accepted whereas the progressive physician might want to explore those care options.

**Relevance of moral distress and ethical climate.** MD has an important impact on healthcare and job resignation through what has been theorised as the crescendo effect. MD, if not properly addressed, gives rise to moral residue [11] or reactive moral distress [12]. Moral residue can lead to anger and frustration. After several morally distressing events, the moral residue increases: the crescendo effect. At a certain point, the health care provider cannot cope with this stress anymore and reaches a breaking point [13]. This breaking point then leads to "ethical stillness": an insensibility to moral distress as a protection mechanism against hurt feelings arising from difficult situations conflicting with own values [14].

Consequently, MD has been linked to individual effects (emotional, physical, spiritual and behavioural), care quality indicators, burn-out and job resignation intentions in health care professionals generally [15–21] and physicians more specifically [5, 22, 23]. Similarly, EC has been linked to job turnover, job satisfaction, patient satisfaction, outcomes, appropriateness of care, and missed care [24–27].

**Towards moral resilience.** More recently, moral resilience (MR) has been coined as a response to negative perceptions of MD. In this view, a MD experience can, in the right circumstances, act as a catalyst, 'canary in the coal mine' or 'tension as springboard for action'

towards positive change [14, 21, 28–31]. Although no single unifying concept of MR exists [32], in this study, we understand MR as a positive response to moral challenges (such as moral distress), thus preventing the crescendo-effect and a breaking point of ethical stillness.

Some authors warn that the concept of moral resilience should not imply unfair responsibility of and pressure on individuals to effectuate positive changes [29]. If contextual constraints cannot possibly be overcome, a caregiver may be unfairly blamed for her or his inability to find a sustainable solution. The sometimes defining role of an underlying professional or institutional culture should not be underestimated [33].

**Moral distress, ethical climate and moral resilience in primary care.** The bulk of studies investigating MD and EC explore intensive, palliative and hospital care settings. Despite some indications that less MD exists in non-intensive care unit settings [24], we have personally heard GP and GP trainee (GPT) colleagues speaking about profound effects of moral distress. As to date however, there is, as far as we have encountered, a paucity of studies explicitly investigating the themes of moral distress and ethical climate in primary care physicians despite increasing burnout numbers [22, 23, 34–36]. With this pilot study we investigate EC and MD in the specific GP context, thus providing GPs with a context-specific framework and insight to enhance moral resilience in their daily practice.

## Research aims

To explore those themes, that have barely been researched in primary care, to gain a background insight and generate hypotheses, we set up this pilot study with small sample size and without exhaustive intent, to investigate the following research questions.

1. How does moral distress occur in general practice?

2. What role does the ethical climate play in the experience of moral distress among GPs?

3. How can ethical climate and moral distress result in moral resilience in general practice?

## Materials & methods

### Study design

A narrative pilot study using a constructivist approach was conducted by the principal investigators (RC and AS), who were in their second year of GP training at the beginning of the study. This approach allowed us to construct a first overview of the research themes based on prior knowledge and theory.

Semi-structured in-depth interviews were taken. Starting point of the interviews was a specific lived example of moral distress. We asked GPs about facts, their thoughts and the competing values in their case. This technique is often used when parsing ethically charged events. We took inspiration from the "First Aid for Ethical Stress" tool developed by a regional hospital in Belgium [37].

From this point, the perspectives were broadened toward ethical climate and moral resilience. Interviews were structured along the three main research questions above and composed based on Mortelmans' "Handbook of qualitative research methods" [38]. We asked questions to probe whether outcomes and determinants named in MD, MR and EC literature (e.g., moral sensitivity, ethical leadership, moral judgment and democratic dialogue, moral motivation, and moral character) could be found in physicians' experiences [3, 7–10, 13, 14, 21, 24, 27, 29, 32, 37–40]. Those questions were based on a prior scoping literature review.

Interview questions were subject to modification based on insights obtained after execution of several interviews in an iterative process involving regular discussions between the main

authors. A revision of the interview guide was conducted after two preliminary interviews, based on experiences and difficulties encountered, thus reducing risk of interviewer bias through relative inexperience of interviewers.

Interview guides can be found in S1 File. Participants were recruited from 18 December 2020 until 21 September 2021; interviews were recorded from 24 April 2021 until 12 October 2021 and took place predominantly through tele-interviews in accordance to hygienic limitations due to the Covid19-pandemic during this timeframe. A minority was recorded in real life or through phone call. Interviews took approximately 1 hour.

## Study population

We invited GPs and GPTs from 2 distinct GP groups in Flanders, near Brussels (Belgium) to participate in our study. The groups consisted of 137 GPs and 135 GPs, 12 GPTs and 14 GPTs at the start of the study, respectively. We chose this target population for convenience reasons (proximity): interviewers RC and AS worked in these groups at the time of study. Consequently, the interviewers knew several of the participants beforehand from professional cooperation. Participants were recruited from a survey distributed through the respective group and medical guard post coordinators, initially with a goal of saturation which was later deemed practically impossible (as mentioned in the discussion section). The sample was purposively extended through personal invitation by e-mail or phone call by RC and AS to acquire a balanced sample considering age, professional experience, practice setting and sex. Written informed consent was obtained prior to each interview.

## Data analysis

During interviews, data were recorded on audiotape after consent of participants. Data were anonymized and stored according to Belgian legislation concerning privacy. Data were processed and analysed based upon methods described in the Qualitative Analysis Guide of Leuven (QUAGOL) [41], which is a commonly used framework for qualitative research. All authors had access to anonymized ad verbatim transcripts of all interviews. We tried to minimize bias by data triangulation. Transcripts of interviews were read and concepts distilled independently by the two principal researchers RC and AS individually and then merged after discussion. A small sample of interviews was read and discussed by junior (RC, AS) and senior (JDL and YD) researchers, thus assuring reproducibility of interview schemes and minimizing researcher bias. In general, we found that conceptual interview schemes did not differ importantly between individual researchers. Disagreements between researchers on conceptual schemes were resolved after discussion. Conceptual interview schemes can be found in S2 File. A fitting test of interview schemes and forwards-backwards movement between interviews and concept schemes was performed as a means of establishing further trustworthiness of findings through triangulating data analysis. This process was debriefed during intervisions between researchers. A final conceptual overarching interview scheme was derived from the data. Coding was done manually. Return of transcripts, codes of themes to participants for comment, correction or feedback was not part of the QUAGOL methodology we used. Here, we report findings in accordance with the Standards for Reporting Qualitative Research (SRQR) [42]. A checklist can be found in S1 Table.

## Ethical approval

This study was part of a broader master's thesis dissertation by RC and AS on MD and EC in primary care physicians. The whole study protocol with reference number MP016596 was approved by the Research Ethics Committee UZ/KU Leuven on December 8th, 2020 and was

found to be in accordance with the ICH-GCP principles, with the most recent version of the Helsinki Declaration and with applicable laws and regulations.

## Results

Thirteen different doctors were interviewed. Characteristics of the different doctors can be found in Table 1.

### Main findings

To facilitate overview, we first provide an overview of main findings responding to the research questions, subsequently providing more detail.

**How does moral distress occur in general practice?.**    GPs report significant emotional, physical and existential impact, high job turnover and quality of care effects when confronted with MD.

Direct causes of moral distress are communication problems, divergent views of what is perceived as "good care" and impending harm to the collective interest. Those direct causes or conflicts can occur between persons, between person and organisation and between organisations.

Micro, meso- and macro-level determinants interact intricately and influence the likelihood of MD occurrence in certain situations and influence the consequences of MD on people.

On a micro- or personal level typologies such as a high sense of duty, an urge to control and high self-expectations are described as negative contributors to moral distress. Professional and life experience as well as a steady work-life balance play a major role at this level.

On a meso-level, apart from the doctor-patient relationship, the whole organisational ethical climate (including dynamics within and practical collaboration) determines the occurrence of moral distress.

On a macro-level, (perceived) scarcity determines moral distress. Relationships between GPs and safe exchanges about moral distress suffer from alienation through scale increases and digitalisation.

**What role does the ethical climate play in the experience of moral distress among GPs?.**   The ethical climate of a GP practice is an important meso-level determinant of moral distress. It can be a source of moral distress; on the other hand, it can also enhance moral

**Table 1. Population characteristics in-depth interview.**

|  | N total (%) |
|---|---|
| **Total number of doctors** | 13 (100) |
| **Self-declared sex** |  |
| Female | 6 (46) |
| Male | 7 (54) |
| **Education** |  |
| General practitioner | 11 (84) |
| General practitioner trainee | 2 (16) |
| **Collaboration** |  |
| Group practice | 7 (54) |
| Duo practice | 1 (8) |
| District health center | 1 (8) |
| Solo doctor | 4 (30) |
| **Mean age (years)** | 44,7 |

resilience. A safe, open space where an authentic and diverse dialogue exploring different perspectives can be lived, is primordial: during our reflexive interviews GPs came to see distressing experiences in a different, more manageable light. This dialogue implies an ethical learning process where GPs develop awareness, vocabulary and skills to address MD.

**How can ethical climate and moral distress result in moral resilience in general practice?.** When confronted with a moral distress experience, general practitioners can bounce back and bounce forward, if the right circumstances are present. Circumstances conducive to resilient ethical climates can be created.

Doctors can bounce back by rebalancing their professional moral distress burden with their carrying capacity through venting distressing events with peers and free time predominantly, reducing work stress and thus not allowing moral distress to gain the upper hand. This bouncing back enabled doctors to take a distance and see things in a different light, consequently working on the way forward ("bouncing forward").

Doctors bounced forward through learning from moral distress events professionally. This learning occurs through self-reflection and exchanging views with colleagues. After taking the first step of bouncing back, taking a distance, things can be seen from a different perspective.

## Moral distress: Individual consequences

Looking at negative consequences of MD in everyday practice in more detail, we found the following, as summarised in Table 2.

Intense emotions, such as guilt, grief, fear, self-doubt, anger, incomprehension and uncertainty were experienced when GPs got stuck.

Following these intense emotions, physical consequences emerged: reduced sleep quality, increased stress and loss of energy.

On the existential level, we saw doctors doubting their own skills and role, for example during the Covid-19 pandemic when GPs weren't allowed to visit their own frail patients in residential care centre:

*"You have to have some control, don't you? Control over your patients. If you have years of experience, you know how to touch people. You do have a feel for it. And then (during the covid pandemic) it didn't work." (D6)*

Avoidance of moral distress situations turned out to be a recurring coping strategy, although quality of care was suffering:

*"If they don't like my ideas, I'm not going to impose them any further. (...) I have indeed learned that it is better to remain silent when I have an idea."(D3)*

Some doctors thought about leaving their jobs because of MD. They nevertheless continued because they saw no other option.

**Table 2. Moral distress, consequences.**

| Emotional | Physical | Existential | Situation avoidance | Job resignation |
|---|---|---|---|---|
| Guilt | Sleep quality | Doubting role as doctor | | |
| Grief | Loss of energy | | | |
| Uncertainty | Stress | | | |
| Fear | | | | |
| Anger | | | | |
| Incomprehension | | | | |

*"If moral distress increased, I would change jobs. If there were no shortage of GPs, I would probably have changed already." (D11)*

## Moral distress: Causes

An overview of moral distress causes can be found in Table 3.

**Differing opinions of "good care".** A major cause of MD was the clash of differing opinions on "good care", in which a consensus cannot be reached. These disagreements occur at various interpersonal levels: between caregivers, between GP and GPT, between GP and patient, as well as between GP and professional organisation.

*"With different parties who were in favour of palliative sedation I did not dare to contradict anymore (...). I sometimes start palliative sedation without fully supporting it."(D2)*

Another example we saw during the covid-19 pandemic. The government decided that GPs were no longer allowed to visit and treat their patients in residential care centres. One of the GPs (D6) testified how it gnawed not being able to care of vulnerable patients in residential care. After all, GPs assumed that they were the most appropriate person to treat these patients. Yet they had to watch as other doctors–who did not know their patients at all–took over the care, sometimes with negative outcomes.

**Accomplice in harm to third parties or the collective interest.** In addition to MD experienced by doctors when their own moral compass cannot be followed, the interviews showed that MD can also be observed among doctors when harm to third parties is imminent: for example, children, caretakers or the collective social interest:

*"In this case we are also the children's family doctor and we don't want the children to be endangered (...) by adults who consciously choose not to deal with their own problems or not to seek or accept help (...) Those children have been suffering for months!" (D10)*

**Communication issues.** Communication problems recurred at the micro-, meso- and macrolevels (cfr. infra) in the interviews. Several examples already demonstrated the close interconnection of communication with other determinants.

*"When arguing, we are often talking about 1 tree instead of the forest as a whole" (D10)*

GPs mentioned incorrect assumptions, digitalisation, language and cultural barriers, unclear communication about priorities and perspectives, patient's inability to communicate through mental illness or dementia as important causes of moral distress.

**Table 3. Moral distress: Causes.**

| Different opinions of "good care" | Accomplice in harm to third parties | Communication issues |
|---|---|---|
| Between fellow health care practitioners | Caregivers | Incorrect assumptions & prejudices |
| Between GP and patient | Children | Induced by digitalisation |
| Unjustified questions | Doctor-patient relationship | Through language barriers |
| Inconsistent patient | Collective societal interests | Through cultural differences |
| Between GP and professional bodies | | Through patient's illness |
| Local | | Mental |
| Supralocal | | Dementia |
| | | Through unclear priorities |

**Table 4. Moral distress: Determinants.**

| Micro-level determinants | Meso-level determinants | Macro-level determinants |
|---|---|---|
| Typologies | Ethical climate | Scarcity |
| Sense of duty & responsibility | Approach to ethical climate determinants | Physical resources |
| Stubbornness | Informal | Staff |
| Drive for control | Proactive & structural | Knowledge |
| Assertiveness | External coaches, training, review | Time |
| Self-confidence | Use of guidelines and vision texts | |
| Personal vision | Structural meetings | |
| Stress resistance | Practical & financial organisation | |
| | Peer relations, dialogue, decision making | |
| | Sensitivity to colleague's wellbeing | |
| | Familiarity | |
| | Trust | |
| | Eagerness to learn & change | |
| | Modesty | |
| | Approachability and availability | |
| | Professional competence | |
| | Balance: like-mindedness and diversity | |
| | Hierarchy and equality | |
| Experience | Doctor-patient relationship | Alienation |
| Life Experience | Shared decision making | Scale increases |
| Professional experience | Respecting doctor's personal convictions | Digitalisation |
| Other life roles | | |
| Family role | | |

## Moral distress: Determinants

Determinants of MD could be found at three levels subsequently called the micro-, meso- and macrolevels, as displayed in Table 4. They interact intricately and influence the likelihood of MD occurrence in certain situations and influence the consequences of MD on people.

**Micro-level.** Several typologies with negative impact on MD were identified by the doctors: a high sense of duty and responsibility, stubbornness, a drive for control and low assertiveness. As positive characteristics–that would perpetuate the handling of moral distress–stress resistance, self-confidence and a strong personal vision were cited.

> *"It is important that this more introverted colleague be actively asked what he thinks, because he will not say it spontaneously. (. . .)" (D12)*

Experience in both professional and non-professional life proved decisive in the perception of MD situations.

> *"We have toiled and struggled and had to work it all out for ourselves. (. . .) but by trial and error you get through it, of course." (D9)*

A lack of experience and knowledge in certain care settings causes insecurity in manifesting an own vision. Doctors recounted how certain crucial life experiences have helped to put things into perspective whereas other experiences have led to avoidance in morally distressing situations. The doctor from the quote below systematically avoided addressing a hierarchic

superior about various situations that went wrong on an ethical level in the institution where she worked.

*"As a young child I once had to show up at the police station unjustly. Ever since I have been afraid of people who are standing above me." (D3)*

Other life roles can threaten the capacity to act according to moral beliefs. When things were not going well at home, the experience of stress was greater at work. The reverse was also mentioned to be true.

*"There are professional tasks I should be completing, but I can't do them, or I'll be late for school to pick up my children. Then I have to choose between my patients and my children." (D11)*

**Meso-level.**   Some doctors indicated that the doctor-patient(-family) relationship is of great importance. Such an existing relationship of trust can both evoke and prevent MD:

*"Once you get to know the patients, it is often more difficult to say no. (…) (On the other hand,) I think that knowing the life story, knowing the patient, seeing the patient often (…) builds up the trust" (D7)*

Then there was the respect from doctor to patient through shared decision-making. Lastly, doctors also asked patients to respect the authenticity of the doctor:

*"Our patients may occasionally realise that we do indeed have our own views and limits, which they should respect." (D10)*

On the meso-level, several determinants of ethical climate are relevant to development and impacts of MD on GPs.

GPs differ in how they approach the meso-level determinants of ethical climate.

A broad spectrum from structural and proactive to informal approaches of ethical climate exists, mostly based on work experiences within a dysfunctional ethical climate.

Some preferred an informal approach through implicit consensus, with a reluctance towards more structural approaches:

*"We have exactly this kind of flow of norms and values that we respect among ourselves (…) The approach remains always personal (…). The nuances are different but our norms and values around, for example, the end of life are perfectly the same (…). And the informal contact between colleagues: we don't work on that; we just do it." (D13)*

Several doctors who had previously faced moral distress due to a dysfunctional ethical climate showed a more proactive approach. In this more proactive, structured approach, team functioning received great attention through refresher courses, performance reviews and coaches. Contact points for certain topics (confidential advisors or managers), structural meetings with a fixed agenda around structural, practical and ethical themes within and among primary care structures were found to be useful. The importance of an explicit mission statement in general practice groups varied. Sometimes it was regularly revised; to others, although it existed, it seemed of minor importance. An overarching mission statement from umbrella organisations for GP medical practice also provided guidance in making some decisions with

ethical connotations. In other interviews, a consensus on evidence-based medicine (EBM) emerged.

Several ethical climate factors exacerbated moral distress in our subjects.

A failing practical organisation within group practices and discussions about finance caused resentment, whereas thorough and timely discussion prevented this:

*"If this (a practice-related decision) is prepared so intensively, I will have fewer problems with it. If you don't want something and suddenly it's forced upon you, it's not going to go well. (D11)*

Most determinants could however be found in peer relationships, the way they influence dialogue and decision-making.

Trusting relationships and familiarity enhanced open dialogue. This building up of trust took place, among other things, through informal contacts and get-togethers (e.g. eating out) and availability in case of personal problems:

*"With the colleagues of (village X) there are no barriers at all to discuss issues. We (. . .) can speak freely about anything. You can see that when we go to the restaurant together."(D8)*

*"I'm not going to be able to bottle up my frustration until our intervision time on Friday."* *(D7)*

A sensitiveness to colleague's wellbeing enhanced this environment of trust and familiarity.

*"In another practice it was important that you worked hard from 8 a.m. to 9 p.m., whereas here one would rather think: oops, is this colleague alright working so late*?" *(D12)*

When dialoguing with colleagues, openness, approachability, an eagerness to learn, modesty, democratic decision-making and a willingness to change were regularly mentioned:

*"I always had good communication and was easily accessible. They could contact me at the drop of a hat and there was always good feedback."* *(D9)*

*"When a colleague explores the matter further, I am interested in it. I am never convinced of my own vision. I am open to another vision and want to explore it. If it turns out to be better, I want to adopt."* *(D8)*

Absence of these factors leads to a (perceived) lack of openness for discussion of distress:

*"I would not feel comfortable sharing that with the group. Some colleagues laugh these things off. There is a kind of macho atmosphere."* *(D2)*

*"I don't know if she (the colleague) would want to question her concept and philosophy of palliative care if she wanted to stay with her thoughts on that (other) decision."* *(D10)*

Competence of colleagues was a requirement for several doctors:

*"(Working with) people with sufficient scientific knowledge. Who are well trained. People who are well informed on medicine."* *(D1)*

A balance needs to be found between like-mindedness and diversity within colleagues, thus enabling authentic and personal care.

*"A group is also formed by different characters, each of whom can mean something. That each of those doctors is a unique person. That you can say, for example, that this is really typical of that doctor (. . .) He is good at that." (D6)*

*"(Care that) is tailored to ourselves as well as to our patients (. . .) The feeling of' I'm working here within the boundaries of the law but also within the boundaries of my own conscience (. . .) is also a way to keep your job." (D10)*

Lastly, the potential negative effects of lack of equality and (perceived) hierarchy cannot be underestimated either. A certain (perceived) hierarchy exists between GPTs and their supervisors. Even when there was a good relationship and open communication between the two parties, according to the GPTs interviewed, there were still reasons why they sometimes felt their opinions are subordinate: the supervisor has final responsibility, the GPT has less experience and knowledge in this area, etc.

*"Still, as a GPT you always remain somewhat dependent on the policy of your supervisor. The supervisor bears some responsibility for the actions of the GPT." (D2)*

**Macro-level.**   The covid-19 crisis revealed how scarcity prevents the desired treatment plan from being achieved. Lack of physical resources, personnel and knowledge were repeatedly cited as important determinants of moral distress: lack of proper oxygen concentrators in nursing homes; lack of protective equipment for doctors and nurses; lack of knowledge about the covid-19 disease, the usefulness of testing children for covid-19; unclear and rapidly changing guidelines due to advancing insights.

Examples of scarcity were also numerous outside the scope of covid-19: perception of less quality of care in nursing homes due to financial pressures, inadequate care for psychiatric patients, unavailability of facilities where caregivers or family can sleep with the patient, lack of time due to administrative barriers, lack of sufficient staff in nursing homes to provide adequate care for palliative patients.

*"The nursing staff in the nursing homes are quick to insist on palliative sedation because this is supposedly the best solution for the patient. I sometimes have the feeling that this is practically the easiest solution and that is what is chosen."(D2)*

Therefore, GPs had less and less room to take time for patients, but also for their colleagues. Their empathy and quality of care seemed to suffer because of time and other constraints.

Maintaining a good relationship with colleagues, as stated earlier, is paramount. However, alienation caused by scale increases and digitalisation seem to cause greater distance between doctors, hindering frank dialogue among colleagues.

*"(. . .) when you are a new GP in a new area there is a barrier to expressing something. Especially via electronic means. (. . .). Actually, we should be able to go for a drink on Friday after work (. . .) You should get to know them. (. . .) But the group is too big and time is too limited." (D5)*

## Moral resilience: Bouncing back and bouncing forward

We found examples of positive responses to moral distress (moral resilience) in both the short and the long term. We saw doctors trying to ensure that moral distress did not gain the upper hand in the short term by trying to restore balance between their ability to bear and their MD

**Table 5. Moral resilience: Bouncing back.**

| Restoring MD bearing capacity | Reducing MD burden |
|---|---|
| Venting moral distress feelings | Venting moral distress feelings |
| Obtaining recognition and acknowledgement for distress | Unburden by colleagues taking over |
| | Unburden through "shared responsibility" |
| Humour | Setting boundaries |
| Leisure activities | Waiting |
| Hobbies | Adjusting expectations |
| Sleep | Acceptance |
| Substance use | |

burden ("bouncing back"), as illustrated in Table 5. Stress management, for example, enables doctors to avoid the crescendo effect. We saw that this bouncing back enabled doctors to take a distance and being able to see things in a different light, consequently working on the way forward long-term ("bouncing forward"), as illustrated in Table 6.

**Bouncing back.** To restore their balance, GPs mentioned being able to communicate and engage in conversation with partners, family members, close colleagues, or patients as important. Venting of feelings with colleagues and help from colleagues countered self-doubt by recognition and acknowledging difficult situations as well as constructive feedback.

> *"In the long run you start thinking: 'I'm just not good enough at it, or: is it normal that I'm under so much stress?' Then when you hear from peers that they also suffer from it, it does help, then I can accept that it is difficult." (D11)*

A venting of feelings needs to take place first. After a discussion of a distressing situation with colleagues, the shared responsibility reduced burden as well. Sometimes colleagues can even take over if MD burden is too high.

> *"If I actively follow up on those difficult situations and discuss them in team, I can also deal with them better, because then we are actively working on it. We have done our best. And that makes it easier if something does go wrong." (D11)*

When a morally distressing situation could not be solved, an individual attitude of waiting and seeing, adjusting one's expectations, acceptance and setting out boundaries were strategies to reduce MD burden:

> *"(. . .) well, it's her life, it's her choice. I can do what's best for her, but if she won't accept it, my ability to help her stops." (D13)*

Humour helped some to put things into perspective.

**Table 6. Moral resilience: Bouncing forward.**

| |
|---|
| Reflection |
| Individually |
| Collectively |
| Building knowledge & awareness about MD |
| Developing a lexicon for dialogue |
| Expanding perspectives and increasing palette of solutions through dialogue |
| Improved quality of care |

Leisure time, hobbies, getting enough sleep or using substances, lastly, are named as strategies to reinforce carrying capacity:

*"We take a drink in the evening to flush out the day. I think that's a normal way of doing things." (D8)*

## Bouncing forward

*"l'Addition des fautes passées, c'est l'expérience." (The addition of past mistakes, is experience.) (D8)*

In the longer term, we saw moral distress experiences and a favourable ethical climate resulting in professional growth, where morally distressing or difficult situations were seen as a learning opportunity for both the individual and the collaboration as a whole:

*"I think that in similar cases that happened afterwards, I tried to attach more importance to my own feelings and values. I didn't want to let them be pushed aside by other parties straight away." (D2)*

This learning process included building up a certain knowledge and awareness of moral distress and ethical climate, allowing one to develop a lexicon for dialogue and to take distance from the experience.

*"Moral distress is a new word for me. Sometimes I find it difficult to put things into words. Maybe I can use this to put those things into words. If you are having a hard time with a particular case, maybe I can start to identify why. Is it because of moral distress? If you know better why you are stressed, it is easier to deal with it, to anticipate it and to talk about it. And it is easier to put it aside." (D12)*

Some doctors indicated that MD stimulated (self-)reflection. This happened spontaneously or structured through, for example SWOT (Strengths, Weaknesses, Opportunities and Threats)-analysis. This reflection resulted in enhanced personal MD coping mechanisms, refining of communication, heightened moral sensitivity and broader perspective.

Sharing also provided the opportunity to engage the talents and know the unique perspectives of other healthcare providers, thus increasing the range of possible solutions and bouncing forward:

*"At the same time there is the thought: 'Is there something I haven't thought of?'; and that's why you also discuss it with colleagues: to be able to see if you can't solve it (the moral distress experience) after all." (D12)*

In the end, a moral distress experience can thus improve quality of care: we saw a doctor address the lack of covid-19 protective equipment in a residential care home at the start of the pandemic, which ultimately resulted in greater safety for residents and staff.

## Discussion

### Main findings

Originally coined in nursing contexts, MD was considered by Jameton as A) the psychological distress of (B) being in a situation in which one is constrained from acting on what one knows

to be right [43]. This definition highlighted nurses' external hindrances for doing what is right, clashing with the authority of the physician treating the patient. In any way, we found that GPs experience MD effects through conflicting visions with a hierarchical superior or external authority, too.

Current understanding of MD has extended beyond the notion of external constraints and highlights personal, interpersonal, and societal determinants [36, 44]. It arises through conflicts within a physician's personal beliefs, values and principles (personal constructs) caused by personal, ethical, moral, contextual, professional and sociocultural factors. How these experiences are processed and reflected on and then integrated into the physicians personal constructs impacts their self-concepts of personhood and identity and can result in MD. Our findings of micro, meso- and macro level determinants of MD in GPs are concordant with these more recent concepts.

It is not surprising that outcomes of MD in GPs resemble those found in other care providers, as underlying mechanisms are similar [5, 13–19, 21].

By investigating how the ethical climate influences MD and MR, we identified malleable factors contributing to ethical climates sustaining MR of the organization and its employees when facing distress. This requires acquisition of vocabulary, skills, and knowledge.

### Suggestions for fostering ethical climates

Readers interested in fostering resilient ethical climates might remain wondering how to acquire the vocabulary, skills and knowledge described above. Here, we suggest several already existing pathways that, from our modest point of view, are worth exploring further.

First, knowledge and education about a basic ethical and moral distress framework can help in taking distance from moral distress experiences, and in managing and communicating about them more proactively. Examples of predominant ethical frameworks are: the care ethics framework [45], narrative ethics [46], and the principle-based approach (autonomy, beneficence, non-maleficence, and justice). While reviewing the literature and taking interviews about MD and EC, we (chief investigators RC and AS) can testify from first-hand experience that we learnt a lot to apply in our personal practice as young and starting GPs.

Second, Balint groups and individual psychotherapy may also help GPs to address internal constraints and MD determinants in the doctor-patient relationship [47, 48].

Third, open and constructive communication can be enhanced through the nonviolent communication method developed by Marshall Rosenberg [49]. This approach to interpersonal communication assumes that conflicts arise when human needs are miscommunicated and helps individuals communicate their needs in a compassionate manner.

Fourth, coaches with GP experience can provide support in optimising a fruitful cooperation by helping individuals and collaborations to develop the necessary collaborative skills [50].

Fifth, The American Association of Critical Care Nurses' (AACCN) "4A's model" is another tool in dealing with moral distress. The A's stand for "Ask, Affirm, Assess and Act". It all begins with self-reflection, where the caregiver learns to see that moral distress is present (ask), then affirms to himself that action is important (affirm), draws up an action plan based on the underlying determinants (assess) and then takes action to restore integrity and authenticity (act) [51].

Finally, we suggest *"reculer pour mieux sauter"* (stepping back to take a leap) or structurally making time for the incorporation of reflection, peer review and seminars. Those reflexive interactions don't only broaden ethical perspectives and language but also allow GPs to reflect about what really matters in care, thus addressing perceived scarcity of resources through

developing "care that is good enough". It could also save time through care efficiency: care that most meets the needs of the parties involved [52]. This concept of "Care ethic efficiency in scarcity" is based on Musto's "Doing the best I can do model" [53]. Reflecting and exchanging about moral distress situations in advance and going through possible scenarios and consequences could furthermore enhance healthcare providers' courageous approach to morally distressing situations in real-life experience [54]. This moral courage can be considered a virtue and skill that can be cultivated in a particular context where awareness about MR and EC is alive.

## Strengths and limitations

Although MD, EC and MR seem to be important factors in the wellbeing of care providers as well as quality of care, this study is one of the first empirical studies shedding light on their interrelationship among general practitioners.

Not much knowledge about MD and EC exists among the GPs we interviewed. Thus, few tools exist to address these topics. Here, we partly unveil how MD, EC and MR interact in primary care, thus supplying a starting point for GPs to gain insight into how MD and EC impact on primary care quality and their worker's job satisfaction, ultimately providing a baseline from which further research towards identifying and resiliently addressing morally distressing situations can develop. Future studies investigating effects of interventions on MD, which are currently lacking, seem paramount.

Apart from these positive elements, there are also some limitations.

First of all, several forms of bias may hamper this research: selection bias, interviewer bias and reporting bias. As GP trainees conducting interviews, we could have been inclined to bias our questions and interpretations during interviews and thus unwillingly exclude or not fully understand MD experiences of more experienced GPs, biasing our report. Also, more experienced GPs might have withheld certain answers or not shared thoughts in order not to "shock" junior doctors. We tried to counter these risks through regular feedback rounds with JDL who is a senior GP and YD, an expert in medical ethics. Both have ample experience with qualitative research.

Second, we did not achieve saturation in multiple aspects due to the broad scope of this study and limited number (n = 13) of participants. However, as the intent of this pilot study was exploratory, further research could provide more exhaustive and detailed information. Our most important finding is that GP's MD arises in and through a specific context. This awareness then stimulates GPs to reflect on their MD experiences, thus allowing GPs to identify and address determinants and generative mechanisms of MD. A more extensive sample would perhaps have led to a more extensive index of MD causes/determinants but not have changed this main conclusion.

Third, in our interviews, little data were found regarding the emergence of a shared vision, worldview or ethical consensus in general practice which is underscored by explicit and implicit assumptions about health and the practice of medicine [55]. We believe that seeing and questioning this shared vision could promote moral resilience and suggest further research in this respect.

Fourth, heterogeneity of definitions of MD, EC and MR in earlier research have permeated our research and interviews, thus potentially reducing clarity in conclusions. Further research would benefit from uniform concepts. Nevertheless, we think an open outlook on those phenomena allowed us to picture a rich image essential to an explorative study generating opportunities and hypotheses for more specific research.

Finally, the external validity of this research must be framed within space and time. We believe a certain cultural homogeneity of GPs in Flanders might hinder transferability of our

findings to other regions within Belgium, Europe, or the world. In different regions and cultures, GPs might have other perceptions on MD, MR and EC.

The study took place during the time span of the Covid-19 pandemic. This pandemic caused, among other things, a shift in perspective from individual clinical ethics to public health ethics [56]. The role and tasks of (general) practitioners were thoroughly shaken up, which had a major impact on moral distress and mental well-being [57, 58]. Our report thus could have put more emphasis on COVID-19 related determinants of MD such as alienation between GPs through social distancing, scarcity during the COVID-pandemic, testing guidelines that were debatable, not being able to care for vulnerable patients, etc. We do not however believe conducting this research outside of the COVID-pandemic would have changed the main conclusions of this research significantly.

## Conclusions

With this study, we aimed to get a broad view on the relationship between MD and EC in primary care physicians through semi-structured in-depth interviews. Moral distress and ethical climate in general practice have important effects on quality of care, professional wellbeing, and job resignation. We identified important MD causes, determinants, and coping strategies at several levels.

This pilot research provides a framework for GPs to understand, further research and proactively address the contexts in which moral distress and ethical climates arise, thus enhancing morally resilient general practice.

## Supporting information

**S1 File. Interview guide.**
(DOCX)

**S2 File. Conceptual interview schemes.**
(DOCX)

**S1 Table. SRQR checklist.**
(XLSX)

## Acknowledgments

All authors who contributed to the work met PLOS One authorship criteria.

## Author Contributions

**Conceptualization:** Raf Coremans, Anton Saerens, Jan De Lepeleire, Yvonne Denier.

**Data curation:** Raf Coremans, Anton Saerens.

**Formal analysis:** Raf Coremans, Anton Saerens.

**Investigation:** Raf Coremans, Anton Saerens, Jan De Lepeleire, Yvonne Denier.

**Methodology:** Raf Coremans, Anton Saerens, Jan De Lepeleire, Yvonne Denier.

**Project administration:** Raf Coremans, Anton Saerens.

**Resources:** Raf Coremans.

**Supervision:** Jan De Lepeleire, Yvonne Denier.

**Validation:** Jan De Lepeleire.

**Visualization:** Raf Coremans, Anton Saerens.

**Writing – original draft:** Raf Coremans, Anton Saerens, Yvonne Denier.

**Writing – review & editing:** Raf Coremans, Anton Saerens, Jan De Lepeleire, Yvonne Denier.

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
