## [Decision Letter · Decision Letter 0]

30 Oct 2023

PONE-D-23-16186From moral distress to resilient ethical climate among general practitioners: fostering awareness. A qualitative pilot studyPLOS ONE

Dear Dr. Saerens, Thank you for submitting your manuscript to PLOS ONE. After careful consideration, we feel that it has merit but does not fully meet PLOS ONE’s publication criteria as it currently stands. Therefore, we invite you to submit a revised version of the manuscript that addresses the points raised during the review process.

1.  Kindly provide a  brief and more relevant justification for focusing on primary care physicians in your study. 2.  Kindly re-evaluate your coding methodology and the resulting themes. Perhaps consider  re-arranging your findings in traditional themes and sub-themes, and then correlate these with your mind-map analysis. Perhaps as a form of method triangulation?3. In your methods section briefly elaborate on issues such as validity, including external validity of the study, attempts to achieve data saturation (or why data saturation could not be achieved). Also briefly describe reliability and trustworthiness of the study data and reflexivity.4. Then, kindly summarize the strengths and limitations of the study in the sections on " Study limitations".5. In your Discussion section, kindly elaborate further to distinguish and better characterize 'moral resilience' and 'moral distress' with particular reference to the findings from this study, in the context of other studies in the literature.6. Kindly provide relevant examples of moral distress among physicians/ primary care physicians as opposed to nursing contexts.7. Kindly address all other issues , comments, and recommendations by the peer-reviewers.

Please submit your revised manuscript by Dec 14 2023 11:59PM. If you will need more time than this to complete your revisions, please reply to this message or contact the journal office at plosone@plos.org. Please include the following items when submitting your revised manuscript:A rebuttal letter that responds to each point raised by the academic editor and reviewer(s). You should upload this letter as a separate file labeled 'Response to Reviewers'.A marked-up copy of your manuscript that highlights changes made to the original version. You should upload this as a separate file labeled 'Revised Manuscript with Track Changes'.An unmarked version of your revised paper without tracked changes. You should upload this as a separate file labeled 'Manuscript'.

We look forward to receiving your revised manuscript.

Kind regards,

Sylvester Chidi Chima, M.D., L.L.M, LLD

Academic Editor

PLOS ONE

Journal Requirements:

2. Please ensure that you refer to Figure 1-7 in your text as, if accepted, production will need this reference to link the reader to the figure.

Reviewers' comments:

Reviewer's Responses to Questions

**Comments to the Author**

1. Is the manuscript technically sound, and do the data support the conclusions?

Reviewer #1: Yes

Reviewer #2: Partly

2. Has the statistical analysis been performed appropriately and rigorously? 

Reviewer #1: N/A

Reviewer #2: N/A

3. Have the authors made all data underlying the findings in their manuscript fully available?

Reviewer #1: Yes

Reviewer #2: No

4. Is the manuscript presented in an intelligible fashion and written in standard English?

Reviewer #1: Yes

Reviewer #2: Yes

5. Review Comments to the Author

Reviewer #1: Dear authors,

Thank you very much for the opportunity to read your paper. This article has the potential to

make a valuable contribution to such an important topic. It was well structured and provided

valuable information on a current and essential subject. However, I have some suggestions.

Introduction

p.4, lines 47-48

"The EC then, logically, is a major contextual determinant in the development of a health professional's MD." How? Can we, authors give some examples?

P.4, lines 57-58

"Consequently, MD has been linked to individual effects (emotional, physical, spiritual and behavioural), care quality indicators, burnout and job leave intentions (8-16)."

Cited most references seem to be related to nursing contexts. The mechanisms or experiences regarding burnout and job leave intention due to moral distress between physicians and nurses can be different. Please describe or add unique experiences among physicians based on previous studies.

p.5, lines 61-64

As the authors described 'moral resilience' on page 5, lines 61-64, its concept has been more highlighted nowadays. However, readers can be curious about the backgrounds and recent studies on moral resilience among healthcare professionals or in the context of healthcare settings. Furthermore, moral resilience is one of the main concepts of the present research. Thus, please refer to them.

Methods

Please describe the validity of the present study, like rigour or trustworthiness.

Results

P21, lines 382-385

"Leisure time, hobbies, getting enough sleep or using substances are named as strategies to reinforce carrying capacity: We take a drink in the evening to flush out the day. I think that's a normal way of doing things."(D8)"

How does this statement support the description of moral resilience? I wonder what the differences are between the concepts of moral resilience and stress management or coping skills to deal with job stress. They must be distinguished.

P20, lines 369-405

Especially for those descriptions on page 20, lines 369-405, based on the citation and descriptions of moral resilience in the present manuscript, several readers can beg to differ from the authors' interpretations of moral resilience. I recommend that the authors should reconsider those interpretations of moral resilience or delete these descriptions.

Discussion

In previous literature, the concepts of professional autonomy and moral courage are the essential factors influencing moral distress. For instance, limited professional autonomy can be one of the organisational constraints at a macro level, and lack of moral courage can be related to how to deal with ethical situations or cope with moral distress. However, it seems that they would not be explored in the present research. I am curious about why and how the authors consider such a gap between the present study and the previous study. Please describe them in the discussion section.

Reviewer #2: Thank you for the opportunity to review this interesting paper on moral distress and ethical climate among primary care physicians in Belgium.

I think the topic is interesting, particularly in the way we manage healthcare delivery after the pandemic started. In the present form, the manuscript is not ready for publication. There are some further clarifications needed for some terms used, conceptualization, methods, findings and implications.

Firstly, I feel you need a better justification for interviewing primary care physicians about this topic.

Secondly, I find the results confusing – mainly because there are too many concepts and sub-themes that seem to overlap on another. The results seem to be presented as a list of codes arranged in mind map themes; it gives the impression the data have not been synthesized, lacking in coherent narrative and context. I think it could be presented in a better way.

There are some limitations to your study including external validity and you need to write something on reflexivity.

Lastly there is an important part in the discussion about what “resilience” means; it can easily be interpreted as “gaslighting”. You can easily blame the person for their moral distress without considering what the organization and health care policies can change to reduce it.

I think a revision can be done easily - the main piece of work is to take another look at your coding and try to present a coherent story. I find the mind map unnecessarily complicated - a simple chart presenting the main themes and subthemes should suffice.

Please see attached a full report.

6. PLOS authors have the option to publish the peer review history of their article (what does this mean?). If published, this will include your full peer review and any attached files.

Reviewer #1: No

Reviewer #2: **Yes: **Dr Richard Ma

---

## [Author Response · Author response to Decision Letter 0]

7 Jan 2024

We have addressed all remarks and additional requirements in our rebuttal letter. We refer to the file named "response to reviewers" for complete response.

---

## [Decision Letter · Decision Letter 1]

15 May 2024

PONE-D-23-16186R1From moral distress to resilient ethical climate among general practitioners: fostering awareness. A qualitative pilot studyPLOS ONE

Dear Dr. Saerens,

Thank you for submitting your manuscript to PLOS ONE. After careful consideration, we feel that it has merit but does not fully meet PLOS ONE’s publication criteria as it currently stands. Therefore, we invite you to submit a revised version of the manuscript that addresses the points raised during the review process.

1. Please check and correct all new comments/queries raised by Reviewer 2 as detailed below:

 "Line 240 - i think you mean "venting" rather than "ventilating". Same in line 475, you use "venting" later in that paragraph.

Line 270 - table 2 - instead of "job leave" do you mean resignation?

Line 446 - instead of "qualitative care" do you mean quality of care?

Line 625 - I think "exploratory" might be a better word."

2. Kindly check and correct any other typographical and grammatical errors in your revised manuscript before resubmission.

We look forward to receiving your revised manuscript.

Kind regards,

Sylvester Chidi Chima, M.D., L.L.M, LLD.

Academic Editor

PLOS ONE

Journal Requirements:

Reviewers' comments:

Reviewer's Responses to Questions

**Comments to the Author**

1. If the authors have adequately addressed your comments raised in a previous round of review and you feel that this manuscript is now acceptable for publication, you may indicate that here to bypass the “Comments to the Author” section, enter your conflict of interest statement in the “Confidential to Editor” section, and submit your "Accept" recommendation.

Reviewer #1: All comments have been addressed

Reviewer #2: All comments have been addressed

2. Is the manuscript technically sound, and do the data support the conclusions?

Reviewer #1: Yes

Reviewer #2: Yes

3. Has the statistical analysis been performed appropriately and rigorously? 

Reviewer #1: Yes

Reviewer #2: N/A

4. Have the authors made all data underlying the findings in their manuscript fully available?

Reviewer #1: Yes

Reviewer #2: Yes

5. Is the manuscript presented in an intelligible fashion and written in standard English?

Reviewer #1: Yes

Reviewer #2: Yes

6. Review Comments to the Author

Reviewer #1: Thank you for your effort. This revised manuscript will contribute to the healthcare academic field.

Reviewer #2: Thank you for your resubmission and rebuttal. I am satisfied the authors have addressed reviewers' comments comprehensively. I think the manuscript is much clearer. Congratulations.

I only have some minor comments:

Line 240 - i think you mean "venting" rather than "ventilating". Same in line 475, you use "venting" later in that paragraph.

Line 270 - table 2 - instead of "job leave" do you mean resignation?

Line 446 - instead of "qualitative care" do you mean quality of care?

Line 625 - I think "exploratory" might be a better word.

7. PLOS authors have the option to publish the peer review history of their article (what does this mean?). If published, this will include your full peer review and any attached files.

Reviewer #1: No

Reviewer #2: **Yes: **Dr Richard Ma

---

## [Author Response · Author response to Decision Letter 1]

6 Jun 2024

Dear reviewers

In response to your new comments, please find our response below. 

As non-native speakers we acknowledge having used some words inappropriately.

 "Line 240 - i think you mean "venting" rather than "ventilating". Same in line 475, you use "venting" later in that paragraph.

• We adapted the wording to your suggestion. This was done on line 240, 475 and in Table 5.

Line 270 - table 2 - instead of "job leave" do you mean resignation?

• We adapted the wording to your suggestion. This was done on line 74, 84, and in Table 2.

Line 446 - instead of "qualitative care" do you mean quality of care?

• We adapted the wording to your suggestion. The wording was adapted on line 450, too.

Line 625 - I think "exploratory" might be a better word."

• We adapted the wording to your suggestion.

Apart from those changes, we addressed some remaining punctuation and wording issues.

We thank you for your time and constructive and thorough feedback during this review process and hope this research will contribute valuable insights to GPs and researchers.

Kind regards,

Anton Saerens, Raf Coremans, Yvonne Denier and Jan De Lepeleire

---

## [Editor Report · Decision Letter 2]

11 Jun 2024

From moral distress to resilient ethical climate among general practitioners: fostering awareness. A qualitative pilot study

PONE-D-23-16186R2

Dear Dr. Saerens,

We’re pleased to inform you that your manuscript has been judged scientifically suitable for publication and will be formally accepted for publication once it meets all outstanding technical requirements.

Kind regards,

Sylvester Chidi Chima, M.D., L.L.M.

Academic Editor

PLOS ONE

---

## [Editor Report · Acceptance letter]

23 Jul 2024

PONE-D-23-16186R2 

PLOS ONE

Dear Dr. Saerens, 

I'm pleased to inform you that your manuscript has been deemed suitable for publication in PLOS ONE. Congratulations! Your manuscript is now being handed over to our production team.

Kind regards, 

on behalf of

Professor Sylvester Chidi Chima 

Academic Editor

PLOS ONE